# Identification of a Virulent Newcastle Disease Virus Strain Isolated from Pigeons (*Columbia livia*) in Northeastern Brazil Using Next-Generation Genome Sequencing

**DOI:** 10.3390/v14071579

**Published:** 2022-07-21

**Authors:** Mylena Ribeiro Pereira, Lais Ceschini Machado, Rodrigo Dias de Oliveira Carvalho, Thaise Yasmine Vasconcelos de Lima Cavalcanti, Givaldo Bom da Silva Filho, Telma de Sousa Lima, Silvio Miguel Castillo Fonseca, Francisco de Assis Leite Souza, Gabriel da Luz Wallau, Fábio de Souza Mendonça, Rafael Freitas de Oliveira Franca

**Affiliations:** 1Laboratory of Animal Diagnosis, Federal Rural University of Pernambuco, Recife 52171-900, PE, Brazil; myribeirop@gmail.com (M.R.P.); givaldo.bom@ufrpe.br (G.B.d.S.F.); telma.sousa@ufrpe.br (T.d.S.L.); silvio.castillo@ufrpe.br (S.M.C.F.); chicoleite@hotmail.com (F.d.A.L.S.); 2Department of Virology and Experimental Therapy, Oswaldo Cruz Foundation—Fiocruz, Recife 54740-465, PE, Brazil; thaiseyasmine@gmail.com; 3Department of Entomology, Oswaldo Cruz Foundation—Fiocruz, Recife 54740-465, PE, Brazil; lais.machado@fiocruz.br (L.C.M.); odrigodoc2@gmail.com (R.D.d.O.C.); gabriel.wallau@cpqam.fiocruz.br (G.d.L.W.)

**Keywords:** avian paramyxovirus 1, Newcastle disease virus, pigeons, domestic birds

## Abstract

Newcastle disease virus (NDV), also known as avian paramyxoviruses 1 (APMV-1) is among the most important viruses infecting avian species. Given its widespread circulation, there is a high risk for the reintroduction of virulent strains into the domestic poultry industry, making the surveillance of wild and domestic birds a crucial process to appropriately respond to novel outbreaks. In the present study, we investigated an outbreak characterized by the identification of sick pigeons in a large municipality in Northeastern Brazil in 2018. The affected pigeons presented neurological signs, including motor incoordination, torticollis, and lethargy. Moribund birds were collected, and through a detailed histopathological analysis we identified severe lymphoplasmacytic meningoencephalitis with perivascular cuffs and gliosis in the central nervous system, and lymphoplasmacytic inflammation in the liver, kidney, and intestine. A total of five pigeons tested positive for NDV, as assessed by rRT-PCR targeted to the M gene. Laboratory virus isolation on Vero E6 cells confirmed infection, after the recovery of infectious NVD from brain and kidney tissues. We next characterized the isolated NDV/pigeon/PE-Brazil/MP003/2018 by next-generation sequencing (NGS). Phylogenetic analysis grouped the virus with other NDV class II isolates from subgenotype VI.2.1.2, including two previous NDV isolates from Brazil in 2014 and 2019. The diversity of aminoacid residues at the fusion F protein cleavage site was analyzed identifying the motif RRQKR↓F, typical of virulent strains. Our results all highlight the importance of virus surveillance in wild and domestic birds, especially given the risk of zoonotic NDV.

## 1. Introduction

Newcastle disease virus (NDV), commonly named avian paramyxoviruses 1 (APMV-1), is among the most important viruses infecting avian species around the world. This virus belongs to the *Paramyxoviridae* family. Members of this family are enveloped, pleomorphic (mostly spherical), with a negative-sense non-segmented RNA [1]. Its genome is about 15–19 kilobases divided into six genes as follows, nucleoprotein (NP), phosphoprotein (P), matrix (M), fusion (F), hemagglutinin-neuraminidase (HN), and RNA polymerase (L), in the following order 3′*-NP-P-M-F-HN-L-*5′ [2,3]. The *Paramyxoviridae* family is divided into four subfamilies as follows: *Avulavirinae, Metaparamyxovirinae, Orthoparamyxovirinae*, and *Rubulavirinae*. Subfamily *Avulavirinae* is further divided into three genera: *Metaavulavirus, Orthoavulavirus*, and *Paraavulavirus*, where the genus *Orthoavulavirus* comprises the most common Paramyxoviruses that have been isolated from avian species [1]. According to serologic studies, avian Paramyxoviruses (AVPMs) have been divided into twelve different serotypes (APMV-1 from APMV-12), based on hemagglutination inhibition (HI) and neuraminidase inhibition (NI) assays [4].

Newcastle disease virus (NDV) is the most characterized APMV [2]. Outbreaks have been reported in several parts of the world from different avian species. Thus, NDV represents a significant threat to the poultry industry, especially in developing countries where poultry represents a significant source of protein for human consumption. NDV is now considered endemic in most areas around the globe presenting the potential to generate large economic losses [4]. Based on phylogenetic analysis, NDV strains are separated into two classes (named class I and class II), which are subsequently subdivided into different genotypes and clades [3,4]. Class I viruses comprise a single genotype, while class II viruses contain at least 21 genetic groups [3,5,6]. The majority of sequenced class I viruses are from viruses of low virulence found in wild birds [3]. On the other hand, class II viruses were originally broken into multiple genotypes representing more virulent viruses, and also low-virulence isolates [7]. Commonly reported natural reservoirs of NDV include waterfowl, cormorants, and pigeons [4]. Although rigorous biosecurity measures are currently adopted by the poultry industry, the risk of reintroduction of viruses into domestic poultry exists and surveillance of wild and domestic birds is crucial.

Having seen that the NDV genotypes are composed of a wide diversity of isolates, it is not surprising that the pathogenicity and virulence of APMVs exhibit a variable degree [8,9]. Of these, APMV-1 (here referred to as NDV) is the most well-characterized serotype, probably as a direct result of the high morbidity, mortality, and economic loss caused by its highly virulent strains. Thus, based on clinical disease severity, NDV strains were originally classified into groups or pathotypes as follows: lentogenic (avirulent), mesogenic (moderately virulent), and velogenic. According to the World Organization for Animal Health (WOAH), virulent NDV isolates are defined as those with intracerebral pathogenicity index values of over 0.7 (defined as the mean observed score per bird as 0 if normal, 1 if sick, or 2 if dead over the eight-day period) and the F protein cleavage site containing the aminoacid motif characteristic of virulence in chickens [10]. Thus, in more detail, it is possible to evaluate the virulence of NDV strains through the analysis of the F protein cleavage site amino acid sequence [11,12], and by the ability of separation of specific cellular proteases to cleave the F protein pathotypes [13,14]. Lentogenic avirulent NDV isolates have fewer basic amino acids present in the F protein cleavage site than either mesogenic or velogenic strains, which have similar cleavage site sequences [11].

Due to the wide circulation of NDV in poultry populations, there is a significant virus genetic diversity and the continuous emergence of novel variants. Thus, to track NDV evolution and genetic diversity, molecular methods (sequencing and phylogenetic analysis) became powerful tools for the characterization of circulating NDV strains, being extremely relevant given the clinical and economic importance of NDV to the poultry industry and the broad use of live Newcastle Disease vaccines worldwide [3]. Here, we aimed to describe an outbreak of NDV detected from free-living pigeons in Northeastern Brazil in 2018. Complete genome characterization of an NDV genotype VI.2.1.2, detected during this outbreak, was carried out.

## 2. Case Report

In January of 2018, from a town square in a central area of the city of Recife in Northeast, Brazil we started to investigate the mortality and morbidity losses of free-living pigeons (*Columba livia*). This study was carried out in strict accordance with recommendations of international and Brazilian ethic guidelines and approved by the Rural Federal University of Pernambuco Animal Use Ethics Committee (CEUA–Comissão de Ética no Uso de Animais/UFRPE/Protocol Number CEUA-6671030221). The investigation of cases started when local residents reported dozens of dead free-living pigeons appearing in different areas of the city. These were mostly concentrated downtown, in common areas of wild bird agglomeration (public squares and parks). During this outbreak, birds presented clinical signs such as opisthotonos, torticollis, incoordination, diarrhea, depression, lethargy, paresis, and the inability to stand and fly (Appendix A). The official health/veterinary authority was notified of the bird’s deaths. During the outbreak investigation, we observed that birds that presented the clinical signs, as described above, died within 24 h. A detailed investigation of five sick birds, collected from this same location was undertaken in the Laboratory of Animal Diagnosis at the Federal Rural University of Pernambuco/UFRPE. Following necropsy, we documented the following gross findings: diffuse enlargement of the pancreas, engorgement of blood vessels in the air sacs, and hepatomegaly associated with an increased hepatic lobular pattern. In the gastrointestinal tract, significant lesions consisted of hyperemia of the mucosa, associated with an accentuated fibrinous exudate inside the proventriculus, cecum, and small intestine. In one pigeon, the small intestinal content was hemorrhagic. Microscopically, significant lesions were observed in the central nervous system (CNS), liver, kidney, and intestine. In the CNS, there was moderate to severe encephalitis with mild to moderate multifocal areas of neuronal necrosis, neuronophagia, neuropil vacuolation, and lymphoplasmacytic perivascular cuffs in the frontal, parietal, and occipital cortex (Figure 1A,B). In these same brain regions, mild to moderate lymphoplasmacytic meningoencephalitis with diffuse moderate to severe gliosis in the gray matter were observed (Figure 1C,D). Moderate demyelination associated with the formation of digestion chambers (axonal degeneration) was noted in the spinal cord of two pigeons. In the liver, moderate multifocal mononuclear hepatitis associated with discrete foci of hemorrhage and hepatocellular degeneration was noted. The renal tubular epithelium exhibited marked degeneration and cell necrosis, accompanied by moderate multifocal interstitial mononuclear nephritis. Additionally, severe focally extensive necrohemorrhagic enteritis was visualized. Other findings included mild multifocal mononuclear myocarditis, pancreatitis and splenic hemosiderosis. These lesions were found in all pigeons and were moderate in most animals.

Next, tissue samples were collected and then forwarded to the Laboratory of Virology and Experimental Therapy from Fundação Oswaldo Cruz/Fiocruz, Aggeu Magalhães Institute for a complete laboratory investigation. Tissue samples from the necropsy were then assessed by a Real Time RT-PCR (rRT-PCR) protocol specifically designed to detect the matrix (M) gene of a broad range of Avian Paramyxoviruses Serotype-1, as described by Hines et al., 2012 [15]. Briefly, tissue fragments were processed for RNA extraction employing a kit for total RNA extraction from tissues (RNeasy Micro Kit-QIAGEN), following manufacturer’s recommendations. RNA integrity was further assessed by resolving the 28S and 18S ribosomal RNA bands using the Agilent 2100 bioanalyzer protocol. For the rRT-PCR assays a volume of 5 μL extracted RNA and 5 μL random primers (PdN6-New England Biolabs) were reverse transcribed employing the GoScript™ Reverse Transcription System (Promega Corporation, Madison, WI, USA), following manufacturer’s instructions. Next, cDNA samples were amplified employing the kit GoTaq^®^ Probe qPCR (Promega Corporation, Madison, WI, USA) in a QuantStudio5^®^ Real-Time PCR system (Thermo Fisher Scientific, Waltham, MA, USA). Ultrapure water was used as a negative control for amplification, cycling parameters used were as follows: GoTaq^®^ DNA Polymerase activation at 95 °C for 2 min followed by 40 cycles at 95 °C for 15 s for primer annealing and extension at 60 °C for 1 min. All of the reactions were performed in duplicate in a final volume of 15 μL and samples with a Ct value < 38 in duplicate wells were considered to be positive for NDV. As shown in Table 1, we detected the amplification of the M gene from NDV in different tissues from five collected birds. More precisely, NDV viral RNA was detected in 19 different samples (organs) from 5 birds. The lower Ct values (suggestive of higher viral load) were observed for the kidney (mean Ct = 35) and brain (mean Ct = 34.62) samples. Brain and kidney samples were also more frequently positive, with a positive rRT-PCR result in all of the five tested birds (Table 1). 

To further confirm the presence of NDV, as detected by the rRT-PCR assay, we performed a virus isolation attempt from selected positive samples. In summary, we selected those with a lower Ct, suggestive of a higher viral load and enough sample quantity. Virus isolation was performed by the inoculation of previously grown Vero E6 cells monolayers with 50 to 100 µL of mechanically dissociated tissue samples supernatants. For this, tissue samples were mechanically dissociated by gently disrupting approximately 50 mg of tissue into Eppendorf-type tubes after the addition of 300–500 µL of minimum essential media (MEM). Disrupted tissues were then centrifuged at 450× *g* for 5 min and supernatants were inoculated on the top of Vero E6 monolayers, cell cultures were then incubated at 37 °C with a CO_2_ 5% atmosphere. Alterations in cell morphology (cytopathic effect), suggestive of virus presence, were monitored daily to a maximum of 7 days after the initial inoculum. Following inoculation of the processed tissue samples on Vero E6 cells, we observed a significant cell death (characterized by extensive cell detachment and morphology change) from cultures inoculated with brain and kidney samples obtained from bird MP003 (Table 1). The cytopathic effect was evident five days after inoculation (Appendix A). After that, cells were harvested, frozen and thawed, and then centrifuged for 10 min at 450× *g*. The supernatants were then transferred to cryotubes and stored at −80 °C until further usage and confirmation of virus presence by rRT-PCR, as described above. Positive virus isolation from brain and kidney tissues from bird MP003 was confirmed by rRT-PCR on cell culture supernatants, confirming the presence of an infectious NDV from the analyzed bird. All of the procedures were carried out in a biosecurity level 3 laboratory (BSL-3) from Fundação Oswaldo Cruz/Fiocruz, Aggeu Magalhães Institute.

Next, we processed brain samples from bird MP003 for next-generation sequencing (NGS), employing a protocol specifically designed for NDV. As previously applied to other viruses [16], our NGS sequencing protocol is based on a multiplex PCR assay designed to enrich clinical samples and low copies genome samples. Briefly, we designed a set of oligonucleotide primer pairs that generate overlapping amplicon products spanning the entire genome of the NDV (primers sequence and genome position are described in Appendix A). Primer design was based on a multiple genome alignment of NDV freely available sequences (GenBank accession numbers MZ458602, JX532092.1, KX236100.2, KJ577585.1) employing the online available tool PrimerScheme (https://primalscheme.com (accessed on 28 April 2022)). This same approach was applied by our group to sequence other viruses, retrieving nearly complete genomes from low copy samples in different settings [17,18]. For NDV genome sequencing assay, total RNA was used for single strand cDNA generation employing random primers (PdN6-New England Biolabs, Ispwich MA, USA) and ProtoScript II Reverse Transcriptase (New England Biolabs), following manufacturer instructions. cDNA were submitted to multiplex PCR with the designed primers. PCR reactions were performed independently for four pools with Q5 Hot Start High-Fidelity DNA Polymerase (New England BioLabs). A total of 35 cycles of 95 °C for 30 s and 65 °C for 15 min were performed on a Veriti^TM^ Thermal Cycler (Invitrogen, Life Technologies, Waltham, MA, USA). PCR products were then quantified using Qubit dsDNA HS Assay Kit (Thermo Fisher Scientific Inc.). Sequencing libraries were prepared with Nextera XT Library Prep Kit (Illumina, San Diego, CA, USA) using 2 ng of cDNA, using standard NexteraXT 24 indexes and following manufacturer′s instructions. Then this sample was multiplexed with other viral samples routinely sequenced in our NGS sequencing facility in a MiSeq Reagent Kit V3 of 150 cycles. Sequencing was performed employing a paired-end strategy, in the MiSeq (Illumina) machine. A total of 2,144,155 raw reads passed quality filtering using Trimmomatic [19] with default parameters in which 2,105,024 paired reads remained. Then a total of 409,003 reads were mapped against an NDV reference genome (*Newcastle disease virus isolate MM19*, *complete genome GenBank JX532092*) using the tool Bowtie 2 version 2.3.5.1 [20]. We obtained a consensus sequence with 13,518 bp length with a genome coverage of 88.98%. Thus, a nearly complete genome was obtained, this sequence was named NDV/pigeon/PE-Brazil/MP003/2018, which was submitted to GenBank under access number OM418380. The nearly complete genome sequence obtained in this work, along with sequences (n = 62) of highly related viruses available at GenBank, were used to construct a phylogenetic tree. Initial tree(s) for the heuristic search were obtained automatically by applying Neighbor-Join and BioNJ algorithms to a matrix of pairwise distances estimated using the Maximum Composite Likelihood (MCL) approach, and then selecting the topology with superior log likelihood value. A discrete Gamma distribution was used to model evolutionary rate differences among sites (5 categories (+*G*, parameter = 0.4264)). The proportion of sites where at least 1 unambiguous base is present in at least 1 sequence for each descendent clade is shown next to each internal node in the tree. This analysis involved 63 nucleotide sequences. There were a total of 1603 positions in the final dataset after the elimination of all positions containing gaps and missing data. Evolutionary analyses were conducted in MEGA11 [21,22]. We observed that the obtained sequence was grouped with other sequences from NDV genotype VI.2.1.2, previously reported in Brazil, Argentina, and Nigeria. The position in the genome is listed in Figure 2 which is characteristic of velogenic strains of NDV and identifies with the genotype VI.2.1.2. The genome phylogenetic analyses displayed a similar topology to those previously isolated in Southern Brazil 2019 and South Brazil 2014 (Figure 3), sharing the highest nucleotide identity with these two sequences from Brazil 2019 (percentage identity 98.82%, GenBank Accession:MZ458602.1) and Brazil 2014 (percentage identity 97.68%, GenBank Accession:KX097024.1).

Given that the fusion (F) protein of NDV is a transmembrane glycoprotein responsible for viral envelope fusion with host cell membranes, in which amino acid variation in its fusion cleavage site dramatically affects membrane fusion, and although NDV virulence is determined by multiple genetic factors, the F protein cleavage site is known as a major determinant [23]. Based on this, we aimed to identify the amino acid sequence of the NDV/pigeon/PE-Brazil/MP003/2018 isolate F protein. We found that the F cleavage amino acid sequence of our isolate lies with the same virulent F protein sequence motif from Southern 2014 and Southeastern 2019 Brazilian isolates. The same amino acid sequence, defined as “^112^RRQKR↓F^117^” (R arginine; K lysine; Q glutamine; F phenylalanine; Arrow cleavage position; numbers residue position in F protein) is shared among these three isolates, being also the most common observed motif from virulent strains [23]. In addition to this, the motif RRQKR↓F differs from other common avirulent fusion cleavage site motifs GRQGR↓L, GKQGR↓L, and ERQER↓L, most commonly found in NDV Class II genotype VI strains (Figure 3), suggesting that this isolate indeed is a virulent strain (based on the criteria utilized by the WOAH).

## 3. Discussion

In this study, we investigated and confirmed through histopathological examination, viral isolation, and molecular analysis the infection by NDV in free-living pigeons (*Columbia livia*) from northeastern Brazil in 2018. Previous reports on the occurrence of NDV in this region were not available, being this the first detection and molecular characterization of NDV in the aforementioned location. During this work, the authorities were notified and mitigation actions were taken to reduce the likelihood of risk of contagion. NDV is one of the most common causes of disease outbreaks in poultry around the world [24]. In Brazil, this disease is exotic, and a possible NDV outbreak would result in huge economic losses due to high morbidity and mortality rates, and as a direct effect of the need to eliminate infected and “close” contact birds, resulting in expressive loss of meat and eggs production and also by imposing trade restrictions.

In 1984 the pigeon variant NDV caused ND in domestic fowl in Great Britain, and through investigation by sanitary authorities, it was shown that the flocks had been fed grain that had been stored in the Liverpool docks which were contaminated by pigeon carcasses and feces [25,26,27,28,29,30]. From 2000 to 2009 in the European Union (EU) member states, NDV virulent for chickens has been detected in wild birds, domesticated pigeons, and poultry. Based on these isolations it was shown the involvement of racing pigeons infected by NDV, formed by the genetic group 4b(VIb) first seen in Europe in 1981 [31]. The Brazilian poultry industry has an important role in the economy of the country and is the largest exporter of chicken meat in the world, thus, both production and profitability of this activity could be severely affected by the occurrence of a virulent NDV outbreak. Based on that, strict surveillance of NDV and other APMVs is crucial for the maintenance of the country’s poultry industry. 

The pigeons in this study presented neurological signs and lesions commonly observed in other Columbiformes infected by NDV [32,33,34,35]. The gross findings were nonspecific, which is consistent with previous reports from the literature. The histopathological lesions observed in the CNS and gastrointestinal tract were similar to several other reports in avian species including, chickens, turkeys, quails, pigeons, and double-crested cormorants, in which non-suppurative encephalitis, necrosis, and microgliosis were the main CNS findings [36,37,38,39,40,41]. We observed extensive tissue damage in the infected birds, and microscopic lesions were observed in the CNS, liver, kidney, and intestines. The most remarkable and consistent histopathologic changes were observed in the brain and kidneys. Significant infiltration of inflammatory cells was observed in many organs. Among the samples analyzed for virus detection by rRT-PCR, brain and kidney tissues showed the lowest Ct values, which is a direct correlation to a higher viral load in these organs. Clearly, positive infectious virus recovery from brain samples is also correlated to its virulence. This is supported by the findings that avirulent strains are restricted to tissues where trypsin-like enzymes are present, such as the respiratory and intestinal tracts, whereas virulent viruses can replicate in a range of tissues and organs, resulting in a systemic infection [42]. Altogether, these data show that the NDV/pigeon/PE-Brazil/MP003/2018 isolate found in this study has a predilection for infected free-living pigeons’ brains and kidneys, indicative of a NDV virulent strain. Clearly, this study has limitations, histopathological findings are nonspecific and not precisely linked to virus antigen staining at these organs. Moreover, we did not discard other co-infections. Although we successfully recovered infectious virus, the small volume of samples available achieved virus isolation from a single bird only.

Our phylogenetic analysis, using 62 complete NDV genome sequences available at GenBank, revealed that the NDV/pigeon/PE-Brazil/MP003/2018 belongs to genotype VI.2.1.2. This virus was closely related to other viruses isolated from pigeons in Argentina, Rio Grande do Sul (southern Brazil) and São Paulo in 2019 (southeastern Brazil) [32,33,35]. Since the genotype VI of NDV is frequently found in pigeons, this strain is commonly named Pigeon Paramyxovirus 1 (PPMV-1) [27]. Usually, PPMV-1 causes only mild disease in poultry, however, several studies show that according to the number of virus passages, its virulence potential increases considerably [43,44,45,46]. Thus, NDV is a hidden threat to the poultry industry, justifying the importance of investigating this virus in suspected cases in free-living pigeons. As recommended by the WOAH a virulent strain must be classified according to its pathogenicity index, usually assessed through in vivo experimental inoculation of chickens and/or chicken embryos or by the presence of well-defined cleavage sites at the viral F protein. By looking at the F protein cleavage site of NDV/pigeon/PE-Brazil/MP003/2018 we grouped this isolate as a virulent strain. Interestingly, by analyzing natural NDV isolates, Wang et al., 2017 [23] classified the F cleavage site into different types, they found that in general, virulent strains contain the motif R/K-R-Q-R/K-R↓F, while avirulent strains contain the motif G/E-K/R-Q-G/E-R↓L (underlined residues vary between different isolates). Here, the sequence at the F cleavage site (RRQKR↓F) classifies this isolate as a virulent strain, also, this is the most common reported amino acid sequence at F cleavage site from Class II viruses.

In South America the notification of transmissible diseases in free-living birds is scarce. However, in Brazil NDV outbreaks in free-living pigeons were registered in different periods and regions of the country. The first NDV outbreak, identified in an urban area, occurred in feral pigeons from a public square in Porto Alegre, Southern Brazil in 2014 [35]. The phylogenetic analysis of this first report classified the virus as an NDV Class II genotype VI. Here, four years apart (2018), we documented another NDV outbreak 3774 km away from its prior identification. Subsequently, in 2019 in Southeast Brazil, a third outbreak was reported in which the identification of an NDV subgenotype VI.2.1.2 virulent strain was found infecting feral pigeons in Sao Paulo city, the largest city in Latin America [32]. From these data, we can infer that NDV is widespread in the Brazilian territory. The alignment of the three subgenotypes recorded from independent outbreaks across the country confirms the circulation of the same NDV subgenotype VI.2.1.2 in the aforementioned regions.

Our study is the first report of an NDV subgenotype VI.2.1.2 in Northeastern Brazil. Clearly, free-living pigeon populations harboring zoonotic pathogens, often inhabiting large urban areas, can pose a serious threat to the general population. In humans, NDV infection typically causes mild, self-limiting symptoms (mainly conjunctivitis), which develops within 24 h of exposure to NDV and resolves within one week [47]. However, in immunosuppressed individuals NDV infection can be lethal. There are at least two documented cases of fatal pneumonia in immunocompromised patients which were associated with direct contact with free-living pigeons and with infection by NDV [48,49], in addition to a case of fatal encephalitis [50]. Thus, given that virulent NDV strains can cause clinical signs in humans, with at least three fatal cases reported, the surveillance of these viruses becomes essential to better prepare the health authorities.

## Figures and Tables

**Figure 1 viruses-14-01579-f001:**
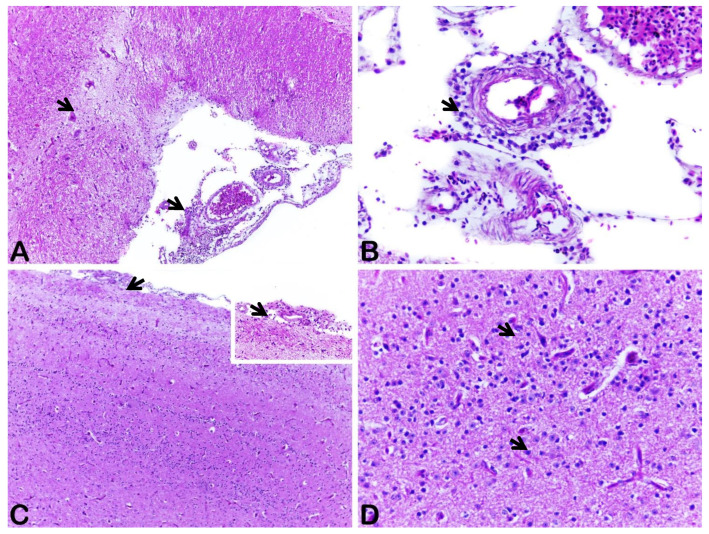
Severe lymphoplasmacytic inflammatory infiltrate expanding the perivascular space and severe vasculitis in leptomeningeal vessels (arrows). Note moderate diffuse vacuolization in the neuropil of the occipital cortex. HE staining, objective magnification 10× (**A**). Blood vessels from the leptomeninges presenting perivascular cuffs are composed mainly of lymphocytes, plasma cells, and rare macrophages (arrows). HE staining, objective magnification 40× (**B**). Mild infiltration of lymphocytes inflammatory cells in the meningeal space and superficial cortex (arrows), mainly composed of lymphocytes, plasma cells, and a few macrophages, diffuse severe gliosis in the inner gray matter. HE staining, objective magnification 10× (**C**). Higher magnification of C where diffuse gliosis and neuronal necrosis are characterized by shrunken, eosinophilic, and pyknotic neurons (arrows). HE staining, objective magnification 40× (**D**).

**Figure 2 viruses-14-01579-f002:**
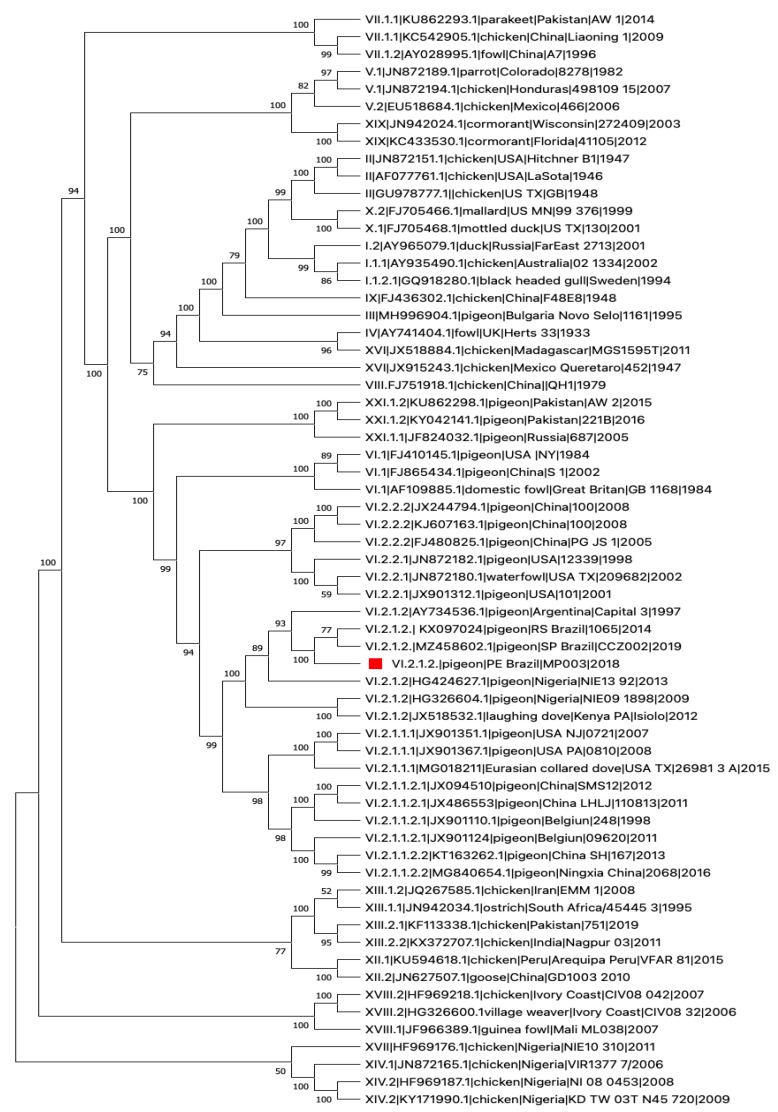
Phylogenetic characterization of NDV/pigeon/PE-Brazil/MP003/2018. Phylogenetic analysis was based on the full-length nucleotide sequence of NDV. The evolutionary history was inferred by using the Maximum Likelihood method and General Time Reversible model. The tree with the highest log likelihood (−18,158.31) is shown. Our analysis classified the NDV/pigeon/PE-Brazil/MP003/2018 as genotype VI.2.1.2 strain (highlighted with a red square). The Roman numerals presented in the taxa names in the phylogenetic tree represent the respective sub/genotype for each isolate, followed by the GenBank Accession number, hostname, country of isolation, and year of isolation (if available).

**Figure 3 viruses-14-01579-f003:**
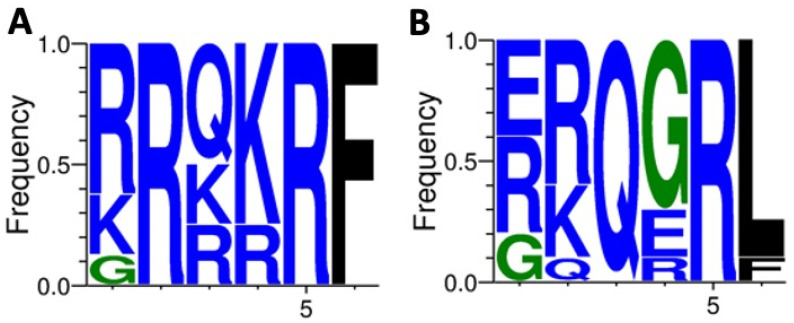
Sequence analysis of F protein cleavage site in NDV isolates. The diversity of residues at each position of the fusion cleavage site was analyzed using WebLogo 3.1 (http://weblogo.threeplusone.com/create.cgi (accessed on 28 April 2022)). Frequency of amino acid residues from the F protein fusion cleavage site motif of virulent NDV natural isolates, representative of 1073 isolates (**A**). Frequency of amino acid residues from the F protein fusion cleavage site motif of avirulent NDV natural isolates, representative of 499 isolates (**B**). The motif ^112^RRQKR↓F^117^ from NDV/pigeon/PE-Brazil/MP003/2018 lies with virulent strains and is the most common motif observed from 801 strains as reported by Wang et al., 2017 [23].

**Table 1 viruses-14-01579-t001:** Description of collected birds and rRT-PCR processed samples. Samples with a Ct (cycle threshold) below 38 were considered positive for APMV-1. H, heart; L, liver; Lu, lungs; S, spleen; P, proventriculus; G, gizzard; I, intestine; AS, air sac; T, trachea; SC, spinal cord; K, kidney; Ct, cycle threshold. * Samples with a Ct result above 38 were considered negative or undetected. ^#^ Not attempted = insufficient sample available to perform virus isolation.

Bird ID	Sample	rRT-PCR (Ct Value)	Undetected rRT-PCR Samples (Ct value > 38) *	Recovery of Infectious Virus
MP001	Kidney	35.0	H, S, G, I, AS,T, SC	
	Brain	34.7		Brain (−)
	Liver	35.9		Kidney (−)
	Proventriculus	36.6		Liver (−)
	Lungs	37.8		
MP002	Kidney	34.8	P, H, S, G, I, AS,T, S	Not attempted ^#^
	Brain	34.7		
	Liver	37		
	Lungs	36		
MP003	Kidney	35.2	Li, P, Lu, H, S, G, I, AS,T, SC	Brain (+) Kidney (+)
	Brain	34.1		
MP004	Kidney	34.9	P, Lu, H, S, G, I, AS,T, SC	Not attempted ^#^
	Brain	35.3		
	Liver	37		
MP005	Kidney	35.1	H, S, G, I, AS,T, SC	Not attempted *
	Brain	34.3		
	Liver	37.2		
	Proventriculus	36.8		
	Lungs	37.4		

## Data Availability

The genome of the NDV/pigeon/PE-Brazil/MP003/2018 is available at GenBank under accession number OM418380.

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
