# Peer review of "Identification of a Virulent Newcastle Disease Virus Strain Isolated from Pigeons (Columbia livia) in Northeastern Brazil Using Next-Generation Genome Sequencing"

_viruses, 2022, doi:10.3390/v14071579_

Round 1
Reviewer 1 Report
The manuscript entitled “Next-Generation Genome Sequencing identifies a pathogenic Newcastle Disease Virus strain isolated from pigeons (Columba livia) in Northeastern Brazil” describes an outbreak of NDV in free-ranging pigeons in Brazil during 2018. The manuscript is well-written and the histopathological lesions are well described. However, the study looks like a case report, does not add new knowledge in the field and lacks comprehensiveness.
Major points:
Genotype VI.2.1.2 has been previously reported in pigeons in Brazil. The neurological changes in the brain of NDV-infected pigeons have also been reported in earlier studies elsewhere.
The presence of multiple basic amino acid residues in the FPCS is suggestive of velogenic nature of the NDV, confirmation by standard pathogenicity testing (ICPI, MDT) are required. No such pathogenicity data has been provided.
The pathogenicity of the isolated PPMV-1 should be studied using experimental infection study and the pathological changes and virus distribution in tissues should be examined to confirm the velogenic nature of the virus.
Other points:
It was not tested (or mention) if the pigeons were free from related co-infections.
L134: Name of the target gene used in the rRT-PCR should be mentioned.
Ct values in tissues are quite high, therefore, presence of viruses in brain should be confirmed by using the IHC or ISH.
L234: Figure legends should indicate the specific lesions in tissues by adding indicators (arrows or asterisks).
Author Response
We thank the reviewer for the suggestions and important comments. We reorganized the entire manuscript to a case report communication format, text is now more concise and we focussed on the findings to report the identification and characterization of NDV. As stated by one of the reviewers figure 3 (WebLogo) is confusing, we are uploading a new figure 3 based on the following: As reported by Wang et al., 2017 the F protein fusion cleavage site (Fcs) sequences of natural isolates are classified into eight types of virulent Fcs (VFcs) with the motif “G/R/K-R-Q/R/K-R/K-R↓F” (these eight VFcs sequences are representative of 1,073 virulent isolates) and ten types of the avirulent Fcs (AFcs) with the motif “G/R/E-R/K/Q-Q-G/E-R↓L” (representative of 499 avirulent isolates). More importantly, the VFcs is only found in the Class II cluster of viral classification and not in Class I. Thus, there is a very limited number of sequences available to compare (at protein level), given the diversity it is also limited. WebLogo figure 3 illustrates the frequency of the following sequence RRQKR↓F from Brazilian virulent isolates versus avirulent strains. The three virulent strains from Brazil share the exact same sequence (RRQKR↓F), which according to the WOAH is enough to classify our isolate as a virulent strain. We then noticed that we should compare the frequency of each amino acid position from virulent strains only, and not from virulent versus avirulent strains as we initially performed. In this revised version of the manuscript we are uploading a novel figure 3, with panels A and B, where Fig3A shows the frequency of each amino acid position among the F protein fusion cleavage site from virulent strains (all the eight of them, as described above) and Fig3B shows the same for the avirulent strains. From this we can conclude that the sequence from NDV/pigeon/PE-Brazil/MP003/2018 is indeed a virulent strain, based on an amino acid motif as analyzed.
Below we will address point by point the comments and suggestions made by this reviewer.
Reviewer #1
Reviewer comment: Genotype VI.2.1.2 has been previously reported in pigeons in Brazil. The neurological changes in the brain of NDV-infected pigeons have also been reported in earlier studies elsewhere. The presence of multiple basic amino acid residues in the FPCS is suggestive of velogenic nature of the NDV, confirmation by standard pathogenicity testing (ICPI, MDT) are required. No such pathogenicity data has been provided. The pathogenicity of the isolated PPMV-1 should be studied using experimental infection study and the pathological changes and virus distribution in tissues should be examined to confirm the velogenic nature of the virus.
Answer: Our findings meet the criteria criteria for virulence of NDV as established by the WOAH, we have not assessed pathogenicity in vivo. We made clear in the manuscript that pathogenicity was not investigated
Reviewer comment: It was not tested (or mention) if the pigeons were free from related co-infections.
Answer: We have not checked for other infections or co-infections, albeit we can not rule out that birds would be carrying other pathogens the lesions observed, clinical signs, PCR result, direct sequencing from clinical samples and recovery of infectious virus particles from the brain and kidney tissues are quite strong data to confirm NDV infection
Reviewer comment: L134: Name of the target gene used in the rRT-PCR should be mentioned.
Answer: A more well detailed materials and methods are provided in the version of the manuscript, target gene is now informed.
Reviewer comment: Ct values in tissues are quite high, therefore, presence of viruses in brain should be confirmed by using the IHC or ISH.
Answer: We agree that Ct values are quite high, however we must consider laboratory variations and use of different protocols, sample aumont, nucleic acid extraction method and many other variables that may result in different Ct values. However, more important, we successfully isolated infectious virus particles from brain and kidney tissues, confirming that NDV was in fact infecting these birds.
Reviewer comment: L234: Figure legends should indicate the specific lesions in tissues by adding indicators (arrows or asterisks).
Answer: We thank the reviewer for this, arrows are now added to the figure to better show the findings from histopathological analysis
Reviewer 2 Report
Authors thoroughly molecularly characterized PPMV-1 from an outbreak in pigeons in Recife city, Brazil, in 2018. The virus clustered with previously reported PPMV-1 in two other regions (South and Southern) of Brazil. The data presented is not a new finding that will make a high impact; however, the case study is clearly described and adds to the knowledge base, showing the detection of this strain in a third Brazilian region (Northeastern). The manuscript should also be revised as a short communication to focus on their findings and avoid repeated information (ex. Introduction, results, and discussion). The authors should also revise the manuscript for minor grammatical editing before publication.
Lines 17-to 19, 63 to 65, 335-339: The authors need to address that not all NDV strains circulating in wild birds threaten the poultry industry, and only a few reports show the reintroduction of the reintroduction virulent NDV (vNDV) from wild to domestic poultry. The authors should address the low risk of transmission of strains adapted to wild birds for domestic poultry based on published experimental studies.
Lines 27, 121, 249, table 1: Four or five pigeons?
Lines 44 to 52. Please update the Newcastle disease virus classification according to the ICTV.
Amarasinghe et al. Archives of Virology (2019) 164:1967–1980 https://doi.org/10.1007/s00705-019-04247-4
Line 52: Abbreviation was already shown in line 38.
Line 77: The World Organization for Animal Health has recently designated his abbreviation as WOAH instead of OIE.
Lines 87 to 97: It would be important to explain better the NDV genotypes and clarify that the PPMV is an NDV genotype.
Lines 111 to 115 or 157: The authors should also state that the virus isolation was carried out according to the regulations, i.e., in a biosafety level 3 laboratory (BSL-3).
Lines 170 to 177. Please clarify the methodology. Explain how you combined random primers and specific primers for NDV using a one-step RT-qPCR kit for sequencing?
Lines 182-185:: I imagine this was multiplexed with other samples at the core facility: specify whether standard nexteraXT indices were used for this. Specify the amount of raw paired reads that were obtained for the sample.d. NGS data analysis: a section clarifying how viral reads were identified completely missing.
Lines 201: Figure 1 shows the histopathology lesions. Please remove this citation here.
Figure 1: Indicate clearly what figure 1 is showing before the letters. Please indicate with arrows or symbols the lesions observed in the tissues of CNS
Line 230: “found” instead of finding.
Line 231: “were” instead of “where”
Figure 3: How many positions were analyzed? Why did you use 200 bootstrap replicates? What was the genetic identity when compared to the identified virus MP003 and the KX097024. How can you explain that MZ458602 (detected in 2019) was grouped closer with KX97024 (detected in 2014) than with MP003 (detected in 2018).
Line 307: Identifying multiple basic amino acids at the C-terminus of the F2 protein and phenylalanine at residue 117, which is the N-terminus of the F1 protein, meets the criteria for virulence of NDV by WOAH. However, the pathogenicity is only demonstrated in vivo. Please consider reviewing the sentence.
Figure 4 is confusing? How many cleavage sites were analyzed using WebLogo 3.1? Did you analyze only genotype VI sequences? Did you analyze only six sequences: three Brazilian and avirulent strains? The authors should explain why they selected the three avirulent strains. Data would be more representative of all genotype VI were analyzed.
figure 4 caption. Line 312: What is the (A)?
Author Response
We thank the reviewer for the suggestions and important comments. We reorganized the entire manuscript to a case report communication format, text is now more concise and we focussed on the findings to report the identification and characterization of NDV. Below we will address point by point the comments and suggestions made by this reviewer.
Reviewer #2
Authors thoroughly molecularly characterized PPMV-1 from an outbreak in pigeons in Recife city, Brazil, in 2018. The virus clustered with previously reported PPMV-1 in two other regions (South and Southern) of Brazil. The data presented is not a new finding that will make a high impact; however, the case study is clearly described and adds to the knowledge base, showing the detection of this strain in a third Brazilian region (Northeastern). The manuscript should also be revised as a short communication to focus on their findings and avoid repeated information (ex. Introduction, results, and discussion). The authors should also revise the manuscript for minor grammatical editing before publication.
Answer: We thank the reviewer for the suggestions and important comments. We reorganized the entire manuscript to a case report communication format, text is now more concise and we focussed on the findings to report the identification and characterization of NDV. Below we will address point by point the comments and suggestions made by this reviewer.
Lines 17-to 19, 63 to 65, 335-339: The authors need to address that not all NDV strains circulating in wild birds threaten the poultry industry, and only a few reports show the reintroduction of the reintroduction virulent NDV (vNDV) from wild to domestic poultry. The authors should address the low risk of transmission of strains adapted to wild birds for domestic poultry based on published experimental studies.
Answer: Most of the literature focuses on describing the potential of virus spreading from wild birds. Our manuscript describes the identification of a virulent strain from a previous outbreak in Northeastern Brazil. Although interesting, experimental studies are less important for the general understanding of our manuscript. Moreover, we could not find the reference the reviewer suggested. If you provide the reference we will be glad to include a few lines in our manuscript.
Lines 27, 121, 249, table 1: Four or five pigeons?
Answer: Issue corrected, five pigeons
Lines 44 to 52. Please update the Newcastle disease virus classification according to the ICTV.
Amarasinghe et al. Archives of Virology (2019) 164:1967–1980 https://doi.org/10.1007/s00705-019-04247-4
Anser: NDV classification is now following the ICTV recommendations, accordingly to Amarasinghe et al. Archives of Virology (2019) 164:1967–1980 https://doi.org/10.1007/s00705-019-04247-4
Line 52: Abbreviation was already shown in line 38.
Answer: Issue corrected
Line 77: The World Organization for Animal Health has recently designated his abbreviation as WOAH instead of OIE.
Answer: Issue corrected
Lines 87 to 97: It would be important to explain better the NDV genotypes and clarify that the PPMV is an NDV genotype.
Answer: Please refer to lines 58 to 64
Lines 111 to 115 or 157: The authors should also state that the virus isolation was carried out according to the regulations, i.e., in a biosafety level 3 laboratory (BSL-3).
Answer: Information added to the manuscript
Lines 170 to 177. Please clarify the methodology. Explain how you combined random primers and specific primers for NDV using a one-step RT-qPCR kit for sequencing?
Answer: Information added to the manuscript, we updated the manuscript adding a better description of the methods employed. Briefly, we reverse transcribed the viral RNA using randon primers and the kit Protoscript RT and the genome was then PCR amplified employing a set of primers designed for the APMV-1 entire genome as described by Quick et al., 2017 (DOI 10.1038/nprot.2017.066), with adaptations as described from lines 203 to 226
Lines 182-185:: I imagine this was multiplexed with other samples at the core facility: specify whether standard nexteraXT indices were used for this. Specify the amount of raw paired reads that were obtained for the sample.d. NGS data analysis: a section clarifying how viral reads were identified completely missing.
Answer: We added a better description of methods employed, lines 26 to 232
Lines 201: Figure 1 shows the histopathology lesions. Please remove this citation here.
Answer: Issue corrected
Figure 1: Indicate clearly what figure 1 is showing before the letters. Please indicate with arrows or symbols the lesions observed in the tissues of CNS
Answer: Information added to the manuscript
Line 230: “found” instead of finding.
Answer: Issue corrected
Line 231: “were” instead of “where”
Answer: Issue corrected
Figure 3: How many positions were analyzed? Why did you use 200 bootstrap replicates? What was the genetic identity when compared to the identified virus MP003 and the KX097024. How can you explain that MZ458602 (detected in 2019) was grouped closer with KX97024 (detected in 2014) than with MP003 (detected in 2018).
Answer: We apologize for this, we noticed an error describing the parameters employed for phylogenetic analysis, this issue was corrected and a complete description of methods employed for tree construction are now corrected in the manuscript, please refer to lines 237 to 246
Line 307: Identifying multiple basic amino acids at the C-terminus of the F2 protein and phenylalanine at residue 117, which is the N-terminus of the F1 protein, meets the criteria for virulence of NDV by WOAH. However, the pathogenicity is only demonstrated in vivo. Please consider reviewing the sentence.
Answer: Issue corrected, we made clear that our findings are suggestive of a virulent strain based on F cleavage site analysis and pathogenicity was not checked in vivo.
Figure 4 is confusing? How many cleavage sites were analyzed using WebLogo 3.1? Did you analyze only genotype VI sequences? Did you analyze only six sequences: three Brazilian and avirulent strains? The authors should explain why they selected the three avirulent strains. Data would be more representative of all genotype VI were analyzed.
Answer: As reported by Wang et al., 2017 the F protein fusion cleavage site (Fcs) sequences of natural isolates are classified into eight types of virulent Fcs (VFcs) with the motif “G/R/K-R-Q/R/K-R/K-R↓F” (these eight VFcs sequences are representative of 1,073 virulent isolates) and ten types of the avirulent Fcs (AFcs) with the motif “G/R/E-R/K/Q-Q-G/E-R↓L” (representative of 499 avirulent isolates). More importantly, the VFcs is only found in the Class II cluster of viral classification and not in Class I. Thus, there is a very limited number of sequences available to compare (at protein level), given the diversity it is also limited. WebLogo figure 3 illustrates the frequency of the following sequence RRQKR↓F from Brazilian virulent isolates versus avirulent strains. The three virulent strains from Brazil share the exact same sequence (RRQKR↓F), which according to the WOAH is enough to classify our isolate as a virulent strain. We then noticed that we should compare the frequency of each amino acid position from virulent strains only, and not from virulent versus avirulent strains as we initially performed. In this revised version of the manuscript we are uploading a novel figure 3, with panels A and B, where Fig3A shows the frequency of each amino acid position among the F protein fusion cleavage site from virulent strains (all the eight of them, as described above) and Fig3B shows the same for the avirulent strains. From this we can conclude that the sequence from NDV/pigeon/PE-Brazil/MP003/2018 is indeed a virulent strain, based on an amino acid motif as analyzed.
figure 4 caption. Line 312: What is the (A)?
Answer: Issue corrected
Reviewer 3 Report
Please check the attachment

Author Response
please refer to file attached

Round 2
Reviewer 1 Report
1. Author should state the novelty of the study
2. Author mentioned that, viral RNA was detected in 19 samples from 5 birds, while virus isolation was successful only from brain and kidney of one bird (MP003). Most of the tissues had an overall similar and high Ct values in the rtqPCR assay. It is not clear if the author attempted virus isolation from the remaining organs of other pigeons. The result of virus isolation should be added in the table 1.
3. Most of the gross and histopathological changes except some changes in the brain are generalized and non-specific. Authors also did not rule out the presence of co-infections in these birds. As the pigeons are free-living, it is not unlikely that these lesions could be secondary. Besides, virus isolation was only performed/successful in two organs (1 pigeon) out of 19 organs (5 pigeons). Lesions were not confirmed upon experimental infection and the virus antigen was not localized in theses tissues having the lesions. Author should qualify the pathological changes and state these limitations in the manuscript.
Author Response
- Author should state the novelty of the study Answer: Thanks for the suggestion, the novelty of this study is limited to the epidemiological molecular characterization of NDV from a location that lacks surveillance of veterinary relevant infectious diseases, as stated from lines 291-292
-
Author mentioned that, viral RNA was detected in 19 samples from 5 birds, while virus isolation was successful only from brain and kidney of one bird (MP003). Most of the tissues had an overall similar and high Ct values in the rtqPCR assay. It is not clear if the author attempted virus isolation from the remaining organs of other pigeons. The result of virus isolation should be added in the table 1. Answer: Virus isolation was performed from samples leftover that were still available (stored), given the small amount of sample from a few birds virus isolation was performed from two birds only, we added this description from lines 177 to 179 and we also added a column at table 1 describing the results of virus recovery.
-
Most of the gross and histopathological changes except some changes in the brain are generalized and non-specific. Authors also did not rule out the presence of co-infections in these birds. As the pigeons are free-living, it is not unlikely that these lesions could be secondary. Besides, virus isolation was only performed/successful in two organs (1 pigeon) out of 19 organs (5 pigeons). Lesions were not confirmed upon experimental infection and the virus antigen was not localized in theses tissues having the lesions. Author should qualify the pathological changes and state these limitations in the manuscript. Answer: Most of the histopathological findings from other reports are also nonspecific, we agree that histopathological alterations would be better explained if accompanied by IHC or antigen detection at these tissues, on the other hand, infectious virus recovery is strong enough to link these findings with NDV infection. As suggested by the reviewer we added a few lines about the limitations of our study (lines 331 to 335)